# Diversity spectrum analysis identifies mutation-specific effects of cancer driver genes

Xiaobao Dong [1]*, Dandan Huang[2], Xianfu Yi[3], Shijie Zhang[4], Zhao Wang[4], Bin Yan[5,6], Pak Chung Sham [6], Kexin Chen[7] & Mulin Jun Li[1,4]*

Mutation-specific effects of cancer driver genes influence drug responses and the success of clinical trials. We reasoned that these effects could unbalance the distribution of each mutation across different cancer types, as a result, the cancer preference can be used to distinguish the effects of the causal mutation. Here, we developed a network-based framework to systematically measure cancer diversity for each driver mutation. We found that half of the driver genes harbor cancer type-specific and pancancer mutations simultaneously, suggesting that the pervasive functional heterogeneity of the mutations from even the same driver gene. We further demonstrated that the specificity of the mutations could influence patient drug responses. Moreover, we observed that diversity was generally increased in advanced tumors. Finally, we scanned potentially novel cancer driver genes based on the diversity spectrum. Diversity spectrum analysis provides a new approach to define driver mutations and optimize off-label clinical trials.

[1] Department of Genetics, School of Basic Medical Sciences, National Clinical Research Center for Cancer, Tianjin Medical University Cancer Institute and Hospital, Tianjin Medical University, Tianjin, China. [2] Department of Biochemistry and Molecular Biology, School of Basic Medical Sciences, Tianjin Medical University, Tianjin, China. [3] School of Biomedical Engineering, Tianjin Medical University, Tianjin, China. [4] Department of Pharmacology, Tianjin Key Laboratory of Inflammation Biology, 2011 Collaborative Innovation Center of Tianjin for Medical Epigenetics, School of Basic Medical Sciences, Tianjin Medical University, Tianjin, China. [5] School of Biomedical Sciences, Department of Anesthesiology, LKS Faculty of Medicine, The University of Hong Kong, Hong Kong, China. [6] Centre of Genomics Sciences, State Key Laboratory of Brain and Cognitive Sciences, The University of Hong Kong, Hong Kong SAR, China. [7] Department of Epidemiology and Biostatistics, Tianjin Key Laboratory of Cancer Prevention and Therapy, National Clinical Research Center for Cancer, Tianjin Medical University Cancer Institute and Hospital, Tianjin Medical University, Tianjin, China. *email: dongxiaobao@tmu.edu.cn; mulinli@connect.hku.hk

Cancer-promoted genetic events and related genes (or so-called driver mutations and driver genes) have been not only successfully identified in most types of cancer but also linked to novel therapeutic opportunities, such as *EGFR* mutations to lung cancer, *BRAF* mutations to melanoma, and *KIT* mutations to gastrointestinal stromal tumors[1,2]. Off-label-targeted therapies, such as NCI-MATCH, aim at treating tumors across anatomical sites based on cancer genomic alterations[3]. However, cancer type-specific and mutation-specific oncogenic signaling has been observed in a number of recent clinical and preclinical studies[4,5]. The quantitative characterization of cancer type preference of driver mutations and their biological and clinical significance remains inadequate.

Mutation-specific effects of driver mutations have been demonstrated in multiple well-characterized cancer driver genes[6–13], which implies that the functional heterogeneities of driver mutations in the same cancer gene could be very common. For example, *NRAS* mutations at codons 12, 13, and 61 were characterized as driver mutations in many cancers. However, only the *NRAS* Q61 mutation can efficiently promote melanoma[9]. Recently, *BRAF* driver mutations were categorized into at least three classes with different kinase activity, RAS dependency, and dimer dependency[6]. More importantly, these mutation-specific effects seem tightly connected with the clinical features of patients. A multicenter clinical study[10] on the efficacy of the HER kinase inhibitor neratinib showed that the responses of patients were determined by both cancer types and mutations, which is consistent with the conclusion of a previous clinical study[14] in which the BRAF inhibitor vemurafenib was tested on patients from different cancer types but harboring *BRAF* V600 mutation. Thus, compared with sophisticated studies at the driver gene level, the development of a unified approach to define the role of each driver mutation will be important to deepen our understanding of cancer genomics and guide clinical trial designs[15,16].

Much work has been done to characterize cancer drivers at a subgene resolution, including at the protein linear sequence, protein domain, protein 3D structure, and protein–protein interface levels[17]. While these methods can provide mutation-level classifications of driver mutations, all of them classify mutations based only on the molecular information of the gene/protein itself and neglect their cancer context, thus may lead to misleading of the effects of mutations. Specifically, the roles of driver genes may vary with different cancer types[18]. Genome-wide screen experiments[19] and a pancancer analysis of the evolutionary selection on driver mutations[20] showed that this phenomenon exists widely. To precisely understand the functions of driver mutations, both the subgene resolution and cancer-context information need to be integrated.

The mutation-specific effects, if they are functional, may unbalance the distribution of each driver mutation in different cancer types, such as *NRAS* Q61R, which is almost exclusively observed in melanoma. Given the cancer distributions of multiple driver mutations from one driver gene, we could distinguish their potential functional differences by comparing their cancer preferences.

In this study, we developed a network-based framework to quantify and compare the cancer preference of driver mutations. By projecting mutations onto a cancer diversity spectrum, we can classify them into three categories, including cancer-specific (SPM), relatively specific (RSM), and pancancer mutations (PCM). The distribution of these mutations in protein domains, genes, and cellular pathways as well as their comutation patterns were systematically characterized. To demonstrate the potential value of the cancer diversity spectrum for clinical and biological problems, we leveraged this information to predict patient drug responses and identify new cancer driver genes. We finally developed a web portal to visualize the cancer diversity for driver mutations at http://mulinlab.org/firework.

## Results

**Network-based measurement of driver mutation specificity**. We first characterized a compendium of driver mutations across 33 TCGA cancer types (see legend of Fig. 1) using more than three million somatic mutations from 10,429 patients. To maximally keep with the conventions of clinical genomic literature and minimize the influence of biased curation in the existing cancer genomics databases, we applied a rule-based approach to identify driver mutations (Supplementary Data 1) in well-characterized cancer driver genes (according to the records of the Cancer Gene Census[18]), which has been widely used in many clinical cancer studies[21,22]. For instance, a missense mutation in an oncogene (OG) would be taken as a driver mutation if it is highly recurrent in cancer patients (recurrence rule). In contrast, a frameshift insertion or damaging missense mutation would be selected as a driver only if this mutation is in a tumor suppressor gene (TSG) (damaging rule).

We constructed a bipartite network (Fig. 1a) to summarize the relationships among patients and 33 cancer types from TCGA project, in which each patient or driver mutation was represented as a node and a patient and a driver mutation were connected if this mutation was detected in the patient. To improve the reliability of subsequent analyses for cancer diversity of mutations, mutations that occur less than three times on the whole TCGA dataset were removed from the network. The final patient–mutation network (Supplementary Fig. 1, Supplementary Files) contains 1570 mutations, 6286 patients (Fig. 1b), and 12,924 edges between them. These mutations belong to 314 cancer driver genes (Fig. 1c), and the highest contribution (16%) is from *TP53*, which is the most frequently mutated gene in cancers[23]. However, there are no individual genes or cancer types that dominate the network.

By compressing all patients from the same cancer type into one node (Fig. 2a), we investigated and visualized the similarity of mutations among all cancer types with force-directed layout algorithm[24]. This algorithm is an intuitive method to spatially organize network data within, usually, a two-dimensional plane. Nodes in the network will repel each other as they were like charged bubbles. On the other hand, each edge will act like a spring to pull a pair of connected nodes together. As the result, cancer types associated with similar driver mutation sets will be clustered and pushed away from other cancer types with different mutation profiles in the final network (Fig. 2a), which allows us to observe the similarity among these cancer types in a globally and flexible manner. The results showed that 79% (26/33) of cancer types shared at least two driver mutations with other cancer types, and 54% (18/33) of cancer types contained at least two private mutations. Cancer types belonging to the same tissues or organs were clustered together, such as two squamous cell carcinomas LUSC and HNSC or two brain cancers GBM and LGG, suggesting that the driver mutation profile can partly reflect the origin of cancers. Few driver mutations were shared with others for relatively rare cancer types, including ACC, CHOL, KICH, PCPG, SARC, THYM, and UVM, which might be attributed to both the small size of the patient cohorts and the distinct molecular characteristics of these cancers, such as the KICH compared with other kidney cancers[25]. Thus, shared and distinct driver mutations composed the patient–mutation networks, which motivated us to precisely quantify the tumor preference of each mutation.

**Specificity-based classification of driver mutations**. We followed a network diversity approach[26] to compute the preference of each mutation (Supplementary Data 2). The network diversity

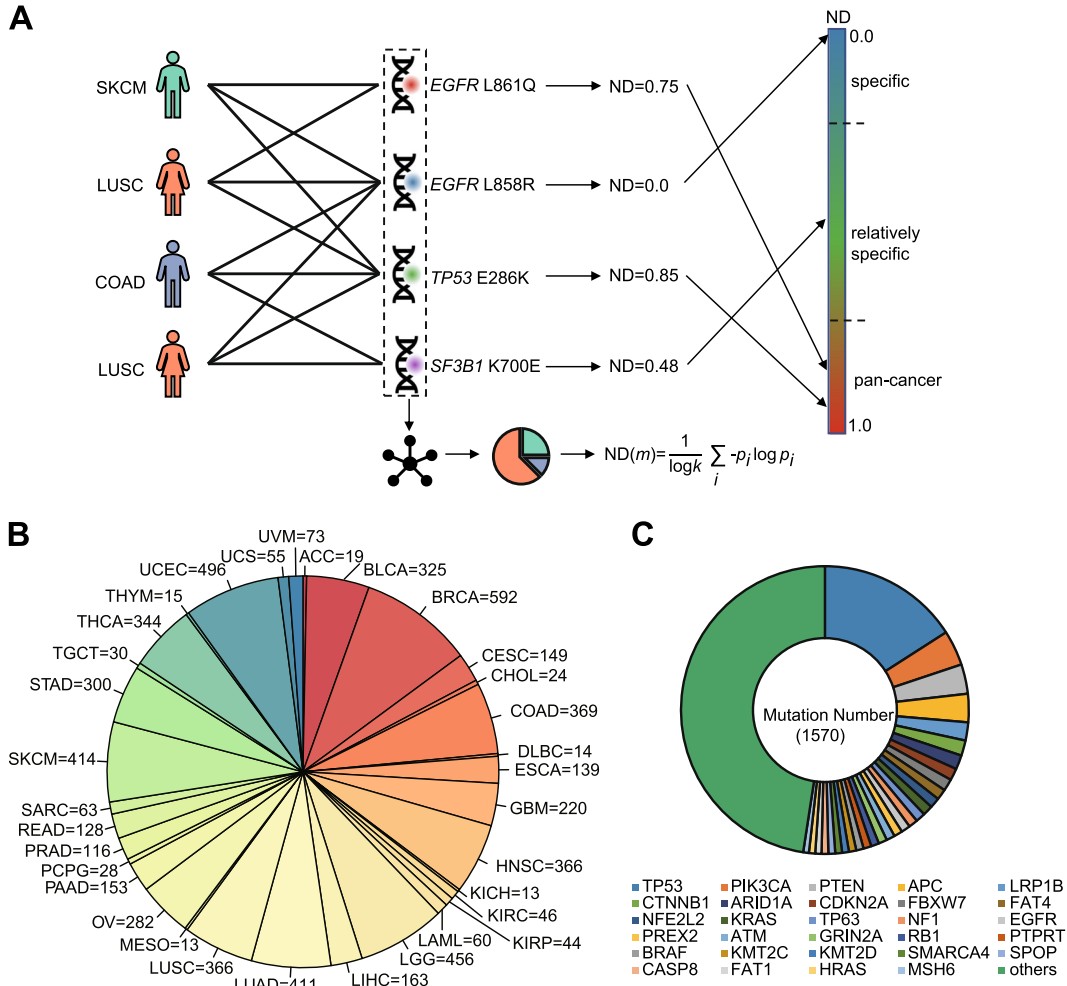

**Fig. 1 Measurement of the cancer distribution of driver mutations with network diversity (network diversity). a** Driver mutations identified from patients of 33 cancer types are used to construct a patient–mutation bipartite network. The 33 cancer types include adrenocortical carcinoma (ACC), bladder urothelial carcinoma (BLCA), breast invasive carcinoma (BRCA), cervical squamous cell carcinoma, and endocervical adenocarcinoma (CESC), cholangiocarcinoma (CHOL), colon adenocarcinoma (COAD), lymphoid neoplasm diffuse large B-cell lymphoma (DLBC), esophageal carcinoma (ESCA), glioblastoma multiforme (GBM), head and neck squamous cell carcinoma (HNSC), kidney chromophobe (KICH), kidney renal clear cell carcinoma (KIRC), kidney renal papillary cell carcinoma (KIRP), acute myeloid leukemia (LAML), brain lower grade glioma (LGG), liver hepatocellular carcinoma (LIHC), lung adenocarcinoma (LUAD), lung squamous cell carcinoma (LUSC), mesothelioma (MESO), ovarian serous cystadenocarcinoma (OV), pancreatic adenocarcinoma (PAAD), pheochromocytoma and paraganglioma (PCPG), prostate adenocarcinoma (PRAD), rectum adenocarcinoma (READ), sarcoma (SARC), skin cutaneous melanoma (SKCM), stomach adenocarcinoma (STAD), testicular germ cell tumors (TGCT), thyroid carcinoma (THCA), thymoma (THYM), uterine corpus endometrial carcinoma (UCEC), uterine carcinosarcoma (UCS), and uveal melanoma (UVM). Based on this network, the network diversity (ND) value of each mutation is calculated and mapped onto the cancer diversity spectrum. According to the spectrum, driver mutations are classified into specific, relatively specific and pancancer mutations. **b** The overall composition of cancer types in the patient–mutation network related to 1570 analyzed driver mutations in the study. **c** The genes that harbor the 1570 mutations and their relative contributions.

is an entropy-based index initially proposed to measure the relationship diversity of an individual in social networks. In our measurement, the network diversity values start from 0 to 1, and a higher value indicates that the mutation is observed in patients of multiple cancer types with a more similar possibility. If a mutation occurs in multiple cancer types and a cancer type dominates the cancer type composition, the network diversity value will be low. On the contrary, if the mutation occurrences among multiple cancer types are similar, the network diversity value will be high. For example, although both *KRAS* G12V and *KRAS* G12R occur in >5 different cancer types, their probabilistic distributions of cancer types are different. There are total 37 patients associated *KRAS* G12R in our data and above 75% of them are PADD patients. In contrast, for the 176 patients associated with *KRAS* G12V, there are three cancer types occupy

much of the composition (23% of PADD, 22% of LUAD, and 19% of COAD). Thus, the network diversity value of it (G12V, network diversity = 0.40) is relatively high than *KRAS* G12R (network diversity = 0.28), representing a different cancer specificity. Note that the network diversity was normalized so that a mutation with high frequency could be compared with a rare mutation directly, which is a merit required for the long-tailed distributed cancer mutation frequency. A continuum of network diversity values formed a cancer diversity spectrum comprising all driver mutations, allowing us to systematically classify and characterize the biological and clinical implications of these mutations.

We found that there are three dominant peaks in the cancer diversity spectrum, which are distributed near network diversity values of 0, 0.5, and 1.0. This trimodal distribution suggests that

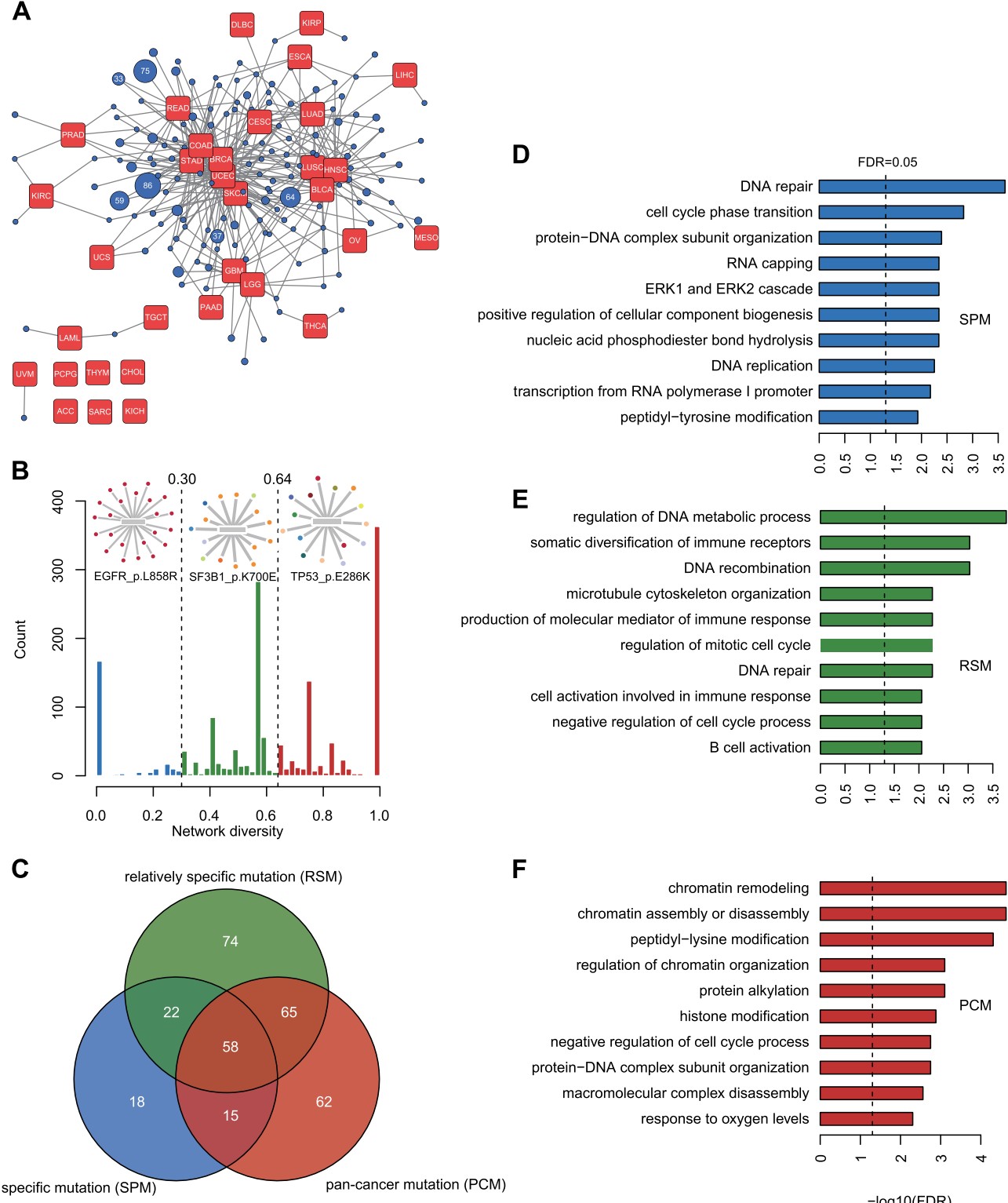

**Fig. 2 Classification of drive mutations and corresponding functional analysis. a** The compressed patient–mutation network in which patients from same cancer types are summarized on a red node. Mutations have same connection pattern with cancer types are compressed into one blue node. The number in a blue node represents the number of mutations included in this node. Note that only node includes at least two mutations are shown. **b** The distribution of network diversity values on cancer diversity spectrum and classification of driver mutations. The mutations above the bar plot are the cases from corresponding categories. Different color nodes connected with a mutation represent patients from different cancer types. **c** The overlap of genes harboring the three types of driver mutations. The GO biological process enrichment results of the SPM (**d**), RSM (**e**), and PCM (**f**) enriched gene network are shown.

driver mutations could be split into three distinct populations (Fig. 2b). Consequently, we classified the mutations into three categories using two theoretically estimated network diversity cutoffs 0.3 and 0.64 (see Methods for details), and generated three types mutations, 230 specific mutations (network diversity <0.3, SPMs), 622 RSMs (0.3 ≤ network diversity < 0.64), and 718 PCMs (network diversity ≥ 0.64). Of note, *APC*, *EGFR*, *PTEN*, *SPOP*, and *LRP1B* are the most frequent driver genes in the SPM category (Supplementary Fig. 2A). This category also includes many known biomarkers for cancer diagnosis or targeted treatment, such as *APC* Q1291* (for COAD), *EGFR* L858R (for LUAD), *BRAF* V600E (for THCA and SKCM), *DNMT3A* R882H and *NPM1* W288Cfs*12 (for LAML). RSMs are exemplified by *SF3B1* K700E, which was mostly observed in BRCA patients (9/15), but sporadic cases were observed in other cancer types (LAML, PRAD, SARC, SKCM and THYM) with low frequency. For this RSM category, *TP53*, *PIK3CA*, *APC*, and *PTEN* mutations were most common (Supplementary Fig. 2B). In contrast to the other two mutation classes, *TP53* mutations significantly dominated the PCM spectrum (Fisher's exact test, *p* value <0.001), which is consistent with a previous integrative study[23] in 12 major cancer types that demonstrated that *TP53* was the only gene mutated near half of the tumors (Supplementary Fig. 2C). Driver genes that harbor multiple types of mutations are common. A total of 18% of genes harbor three types of mutations and 50% of genes harbor at least two types of mutations (Fig. 2c). Except for *TP53* (Fisher's exact test, *p* value <0.01, *q* < 0.01, Benjamini & Hochberg correction), there was no other driver gene significantly enriched in any specific category after multiple hypothesis correction (Supplementary Data 3). Thus, the functional heterogeneity of the mutations could be a common phenomenon from even the same cancer driver gene. More details about the associated cancer types of each mutation can be found in our web portal or Supplementary Data.

To explore biological pathways involved in different categories, we constructed gene subnetworks by mapping the enriched genes of each category onto protein functional networks using the STRING database[27] and performed a Gene Ontology (GO) enrichment analysis to nominate related pathways or biological processes (Fig. 2d–f, Supplementary Fig. 3, Supplementary Data 4–6). The functional analysis showed that DNA repair and cell cycle processes were generally observed in all three categories. However, some processes were specific, including signaling transduction processes, such as the ERK cascade and peptidyl-tyrosine modification, which are mainly enriched in the SPM gene network. Immune response genes are only enriched in the RSM gene network, and chromatin remodeling is the most prominent process for the PCM gene network. These results suggest that certain biological pathways could influence tumorigenesis in specific tissues, while some pathways, such as epigenetic processes, might have a wide impact on tumorigenesis across many cancer types.

**Cancer diversity spectrum and patients' drug responses**. Presumably, even if one driver gene contains multiple driver mutations with varied specificities, then these mutations should appear in separate protein domains corresponding to their specificity categories. To test this hypothesis, we annotated driver mutations in the functional protein domains of the driver gene by using the Uniprot database[28]. Although some domains were enriched with driver mutations, we unexpectedly found that the majority of them harbored more than two types of mutations in the same region (Fig. 3a). A typical example is the protein kinase domain of BRAF protein. In this domain, S467L, G469V, V600M, V600G, and V600E are SPMs, but K601E, G466E, G466V, G469R,

G469A, N581S, and D594N are RSMs or PCMs. One possible explanation is that the annotations of the protein domain are either incomplete or inaccurate. However, to reject the previous hypothesis, we have to explain why mutations located at the same position could belong to different categories, as exemplified by the *BRAF* mutations G469V (SPM), G469R (RSM), and G469A (PCM) and the *KRAS* mutations G12C (SPM), G12R (SPM), G12D (RSM), and G12V (RSM). Previous biochemical studies on *BRAF* and *SPOP* mutations showed that driver mutations could induce very different biochemical behaviors of a protein and exhibit opposite pharmaceutical effects, although these mutations were very closed in linear sequence[6,11]. Our analysis also revealed that cancer diversity classification could distinguish drug response-related mutation effects in the same protein domain. For example, *BRAF* mutations that were sensitive to vemurafenib were classified as SPMs (V600M and V600E), and insensitive mutations were classified as RSMs or PCMs (G469A, G469R, G466V, G466E, N581S, D594N, and K601E)[6]. Similar to vemurafenib, *SPOP* mutations showed BET inhibitor sensitivities that were also consistent with our network diversity-based classifications but in a reverse relationship. Ishikawa cells over-expressing the SPMs of *SPOP*, including Y87C, W131G, and F133L, were resistant to treatment with the BET inhibitor JQ1, while RSMs (R121Q and D140N) were sensitive[11].

To comprehensively investigate the association between the cancer diversity of mutations and antineoplastic therapy, we integrated the cancer diversity spectrum with drug response data predicted by an imputed drug-wide association study (IDWAS)[29]. IDWAS learned statistical models from cell line-based drug response data and gene expression profiles to predict 138 cancer drug responses for 5548 TCGA patients, which allows us to analyze the relationships of the cancer diversity of mutations and drug responses in an unbiased manner. Moreover, IDWAS only uses gene expression data, and its results are independent of gene mutation information.

We evaluated whether there were different drug responses among patients harboring SPMs, RSMs, and PCMs in the same drug target (see Methods for details). Note that because the drug response data from IDWAS are predicted from a gene expression-based statistical model, the drug response values from IDWAS have no clearly defined biological meaning and are not directly comparable with traditional drug sensitivity values such as $IC_{50}$ (drug concentration that reduces cell viability by 50%); however, lower value means greater drug sensitivity. In approximately one-third of the tested drug–gene pairs (30/89), the drug response seemed influenced by the cancer diversity of mutations (ANOVA, $p < 0.2$, Supplementary Data 7), such as temsirolimus-*BRAF* (ANOVA, $p = 0.005$), afatinib-*EGFR* (ANOVA, $p = 2.92 \times 10^{-8}$), gemcitabine-*KRAS* (ANOVA, $p = 0.0009$), and AZD6482-*PTEN* (ANOVA, $p = 0.118$) (Fig. 3b). We also observed that drug sensitivity decreased as the cancer diversity of mutation increased in multiple cases. For example, patients with SPMs of *KRAS* were sensitive to gemcitabine, but the resistance was shown in patients with RSMs and PCMs. The same trend was observed in *EGFR*-mutated patients to erlotinib, *BRAF*-mutated patients to PLX4720, and *PTEN*-mutated patients to AZD6482. One exception is paclitaxel-*KRAS*, in which the drug sensitivity increased with mutation cancer diversity. When compared with the mutation-negative group (i.e., patients who did not harbor driver mutations on the corresponding drug target), the largest number of significantly differential drug responses (two-sided *t*-test, $p < 0.05$) were from SPMs, which were nearly twice or more than the observed number from RSMs or PCMs (Fig. 3c). We also overlapped driver mutations with actionable mutations collected from OncoKB[30] and found that a majority of the actionable mutations belonged to SPMs (Fig. 3d, Supplementary Data 8). Overall, our results suggest that cancer diversity of mutations,

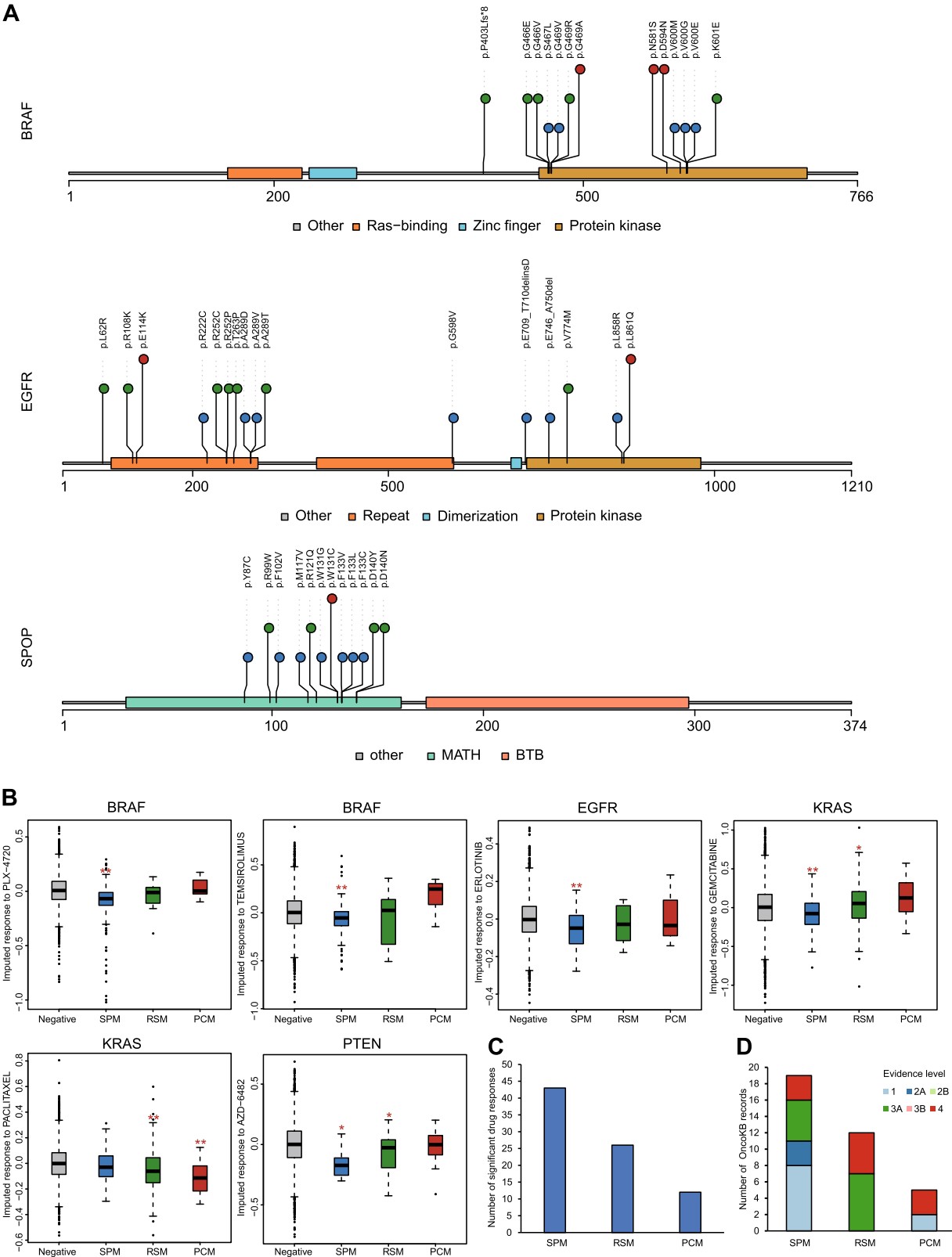

**Fig. 3 Distribution of three types of driver mutations in functional protein domains and the association of cancer diversity and drug sensitivity. a** The distribution of driver mutations in the functional domains of three representative genes. The functional protein domains are annotated according to Uniport records. Three types of driver mutations were distinguished by the color and height of the dots in the lollipop plots. SPMs (blue and short), RSMs (green and middle height), and PCMs (red and high). **b** The drug sensitivity of patients harboring SPMs, RSMs, and PCMs, respectively. The red stars mark statistically significant groups when compared with corresponding negative groups (*$p < 0.05$, **$p < 0.01$, two-sided $t$-test). Drug sensitivity is predicted by IDWAS. **c** The number of drug-mutation combinations that are significantly associated with drug response. Drug sensitivity data are from IDWAS. **d** The composition of OncoKB evidence level in three types of mutations. From levels 1 to 4, the strength of evidence for clinical recommendation gradually decreased.

especially for SPMs, are more correlated with patient drug responses, and such effects cannot be readily inferred from the functional domains of mutations.

**Cancer diversity spectrum and cancer evolution**. To understand the impact of cancer evolution on the cancer diversity spectrum, we first examined the correlation between cancer diversity spectrum and variant allele frequency (VAF) of driver mutations. VAF represents the burden of mutations in a patient and is used as an agent to quantify the relative size of tumor clones harboring certain driver mutations. A high VAF value in primary tumors usually implies that the corresponding mutation was from an early/founder clone. To exclude the confounders that might distort VAF, we selected tumors with cancer cell purity >70% and mutation data from copy number neutral regions. We computed the Pearson correlation coefficient (PCC) between mutations' VAFs and network diversity values for genes with ten or more mutations (Fig. 4a, Supplementary Fig. 4). Among significant correlations, cancer diversity of mutations negatively correlates with VAFs in *BRAF*, *KIT*, *PREX2*, *NRAS*, and *SF3B1* but positively correlates with VAFs in *FBXW7*, *KMT2D*, *NF1*, and *SPOP*. After examining the mode of action of these driver genes, we found that OGs involved more negative correlation relationships, and TSGs included more positive correlation relationships (Fig. 4b). The average PCC values for OGs and TSGs are −0.08 and 0.01, respectively, but the difference between them is not significant (Wilcoxon sum-rank test, $p$ value = 0.079). Considering that high VAF generally indicates an early tumor clone, our results imply that a part of OG-related SPMs and tumor suppressor-related PCMs tend to occur in the early stage of tumorigenesis.

To explore the pattern of different mutation types in the long-term cancer evolution, we compared the network diversity values of driver mutations between primary and advanced tumors. We used MSK-IMPACT data[31] that include genetic aberrations of approximately 400 cancer-related genes from more than 10,000 patients with advanced tumors, representing the mutation landscape of the late stage of tumorigenesis. The mutational frequencies of genes in TCGA and MSK-IMPACT cohorts are highly consistent[31]. We calculated and compared the network diversity values of 625 common driver mutations between the TCGA and MSK-IMPACT groups (Fig. 4c, Supplementary Data 9). The results showed that 57% (359/625) of the cancer diversity classifications of driver mutations were conserved. Nevertheless, 140 RSMs in TCGA increase their cancer diversity and covert to PCMs in MSK-IMPACT (Fig. 4d). Overall, the cancer diversity of mutations in advanced tumors was significantly higher than those in primary tumors (Fig. 4e). Interestingly, we found three mutations, *EGFR* L861Q, *MAP2K4* S184L, and *TP53* E285V, that were PCMs in TCGA but became SPMs in MSK-IMPACT tumors, suggesting that cancer-specific selection may drive them during the continuous progression of related tumors. A previous study related *EGFR* L861Q to the resistance of EGFR-TKI therapy in lung cancer[7], which suggests that this improved cancer specificity in advanced tumors might be attributed to the result of selection during targeted cancer therapies. Taken together, the cancer diversity results of driver mutations not only can influence clonal evolution but also can be reshaped in cancer progression.

**Comutation patterns between mutations from different classes.** It has been demonstrated that there are complex dependencies among driver mutations and that they are related to clonal evolution and the clinical prognosis of tumors[32]. We asked whether there are unique dependencies in mutations with different cancer

type specificities. To answer this question, we performed comutation analysis for all driver mutation pairs and constructed a comutation network that represented significantly co-occurrent or mutually exclusive pairs in all cancer patients (see Methods for details). This network included 1136 interactions among 425 driver mutations, and only 11 of them were mutually exclusive interactions (Supplementary Data 10). To inspect the global dependency between different mutation categories, we created 10,000 random comutation networks by permuting network node assignments as a control distribution. Compared with random networks, SPM–SPM interactions were significantly higher in the observed network (79 vs 41, $q = 0.03$, Benjamini & Hochberg correction, Fig. 4f). In contrast, SPM–PCM interactions were depleted in the observed network (56 vs 120, $q = 0.0006$, Benjamini & Hochberg correction, Fig. 4f). These results suggest that SPMs tend to be comutated with each other but independent of PCMs. We hence inferred that SPMs might be involved in cancer-specific signaling, which was convergently selected in cancer progression. Nevertheless, the retention of PCMs during cancer clonal evolution could be independent of particular tissue or cancer type. Taken together, the cancer type-dependent coevolution of driver mutations may frequently exist in tumorigenesis and could further shape their specificity.

**Searching potential cancer driver genes**. Considering the importance of cancer-specific mutations in cancer evolution and treatment, we asked whether these mutations could be used to predict novel cancer driver genes. Because of the relatively small sample size of one particular cancer type in the TCGA cohort, we scanned potentially new cancer driver genes by examining the distribution of SPMs at the pancancer perspective (Supplementary Fig. 5). First, we expanded our network diversity computation to all protein-coding genes and obtained 8859 SPMs (Supplementary Data 11). Then, these SPMs were taken as input for the MutSigCV algorithm[32] to identify significantly mutated genes compared with background mutation rates. The above procedures resulted in 185 significant genes with $q < 0.001$ (Supplementary Data 12). A total of 45 genes had been recorded as cancer genes in the Cancer Gene Census database, including *BCL9L*, *SFPQ*, *PTPRT*, *UBR5*, and *PAX3* that were missed in a recent TCGA pancancer analysis[33] that combines the genomic data for 33 cancer types with 26 existing bioinformatics tools, which highlights the validity and uniqueness of our method. To further reduce false positives, we filtered out MutSigCV genes without any dependent partners in the cancer dependency map[34], which identified 769 gene dependencies on 501 cancer cell lines by modeling the off-target effects of systematic RNAi experiments. We finally obtained eight bona fide cancer driver genes, including *CEP57*, *HNRNPL*, *KLF5*, *OXA1L*, *PAFAH1B1*, *RBM39*, *SYT13*, and *TFDP1* (Table 1). The gene expression data from patients showed that the expression levels of these predicted cancer driver genes were significantly correlated with patient prognosis in at least one cancer type (Table 1), suggesting the functional importance of these genes in cancers. In addition, we noticed seven of them have been included in the latest version of The Network of Cancer Genes[35] (*KLF5*, *OXA1L*, *RBM39*, and *TFDP1*) or CancerMine[36] (*HNRNPL*, *KLF5*, *PAFAH1B1*, *SYT13*, *TFDP1*), two manually curated cancer gene databases, further confirming our findings.

Among the most representative candidate cancer driver genes are *HNRNPL*, *KLF5*, and *TFDP1* (Fig. 5a). The RNA splicing factor HNRNPL is involved in the repression or activation of exon inclusion in targeted genes, and its high expression consistently correlated with unfavorable prognosis in renal cancer ($p < 0.001$), liver cancer ($p < 0.001$), and pancreatic cancer ($p <$

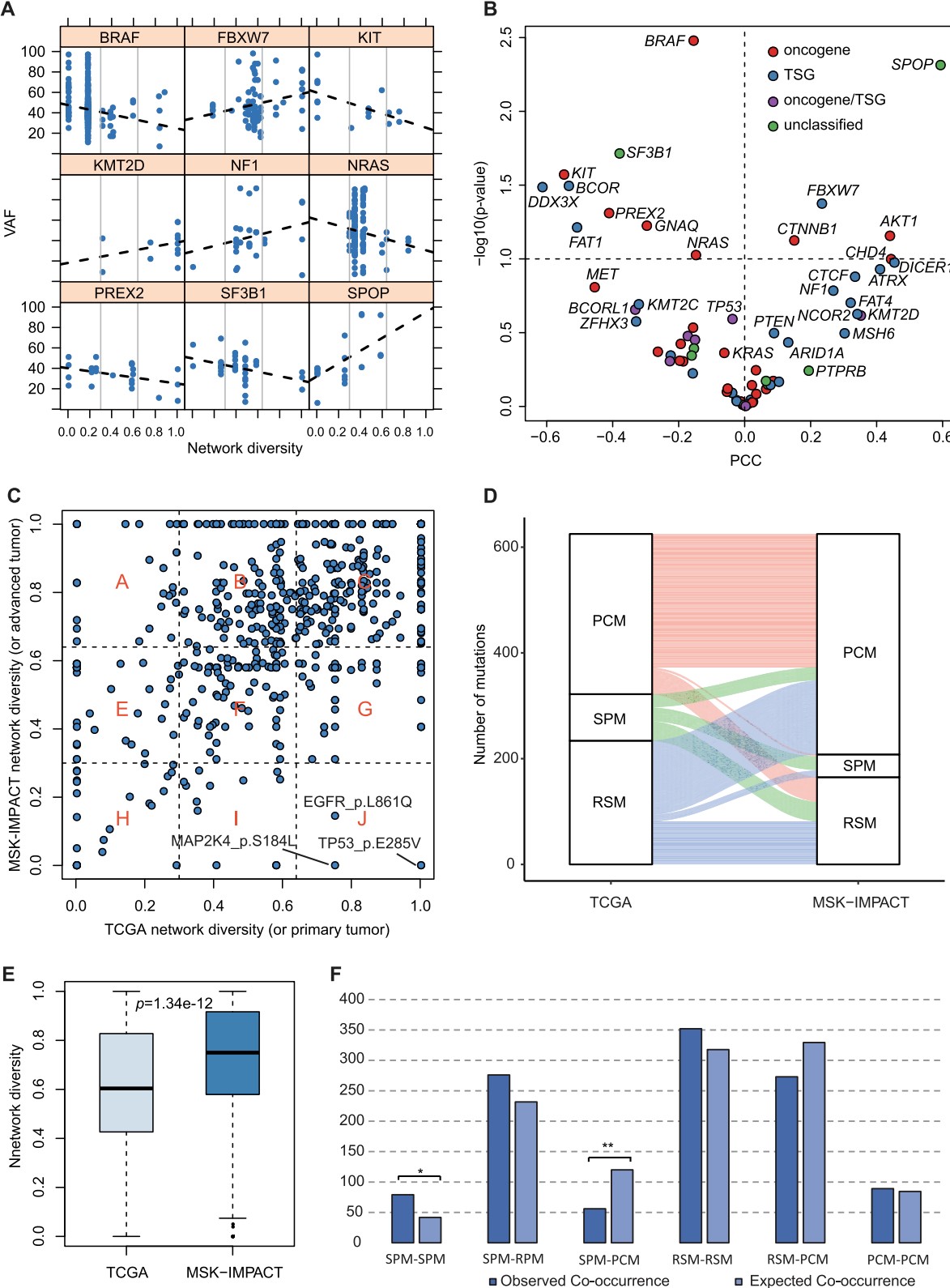

**Fig. 4 Cancer diversity driver mutations and cancer evolution. a** Correlation of VAF and network diversity in nine representative genes. Each point represents a TCGA patient. **b** The overall view of PCCs of VAF and network diversity for genes with different modes of action in cancer. **c** The comparison of network diversity values in primary tumors (i.e., TCGA group) and advanced tumors (i.e., MSK-IMPACT group). The scatter plot is divided into nine regions corresponding to nine combinations of cancer diversity. **d** Sankey diagram illustrates the relative flow of cancer diversity changes from primary (TCGA) to advanced (MSK-IMPACT) tumors. **e** Comparing the distribution of network diversity values between TCGA and MSK-IMPACT. A one-sided *t*-test was used. **f** Observed comutations and expected cases in the different combinations of driver mutations.

**Table 1 Eight potential cancer driver genes identified through analyzing cancer-specific mutations.**

| Gene | Full name | Molecular function | Cancer dependency | Expression and prognosis |
|---|---|---|---|---|
| CEP57 | Centrosomal protein 57 | Centrosomal protein which may be required for microtubule attachment to centrosomes | Esophageal, gastric | Glioma (favorable) |
| HNRNPL | Heterogeneous nuclear ribonucleoprotein L | Splicing factor binding to exonic or intronic sites and acting as either an activator or repressor of exon inclusion | Gastric, pancreas, leukemia, endometrial | Renal cancer (unfavorable), pancreatic cancer (unfavorable), liver cancer (unfavorable) |
| KLF5 | Kruppel Like Factor 5 | Transcription factor that binds to GC box promoter elements | Colon | Pancreatic cancer (unfavorable) |
| OXA1L | Oxidase (Cytochrome C) Assembly 1-Like | Essential for the activity and assembly of cytochrome oxidase | Ovarian, lung NSCLC | Renal cancer (favorable) |
| PAFAH1B1 | Platelet activating factor acetylhydrolase 1b regulatory subunit 1 | Required for proper activation of Rho GTPases and actin polymerization at the leading edge of locomoting cerebellar neurons and postmigratory hippocampal neurons in response to calcium influx triggered via NMDA receptors | Gastric, ovarian, esophageal, colon, breast, leukemia, bladder, lung NSCLC, GBM, endometrial | Renal cancer (favorable) |
| RBM39 | RNA-binding motif protein 39 | Transcriptional coactivator for steroid nuclear receptors ESR1/ER-alpha and ESR2/ER-beta, and JUN/AP-1 | Esophageal, gastric | Renal cancer (unfavorable) |
| SYT13 | Synaptotagmin 13 | May be involved in transport vesicle docking to the plasma membrane | Colon, ovarian | Endometrial cancer (unfavorable) |
| TFDP1 | Transcription factor Dp-1 | Can stimulate E2F-dependent transcription | Lung NSCLC, breast | Renal cancer (favorable), stomach cancer (favorable), liver cancer (unfavorable) |

0.001). HNRNPL P337Hfs*58 is a highly specific mutation in COAD and may interrupt the last two RNA recognition motifs of this protein. Extensive studies on the mutation effects of splicing factors SF3B1, U2AF1, and SRSF2 have demonstrated their altered protein functions in the development of cancers[37]. We hence postulate that a similar mechanism could be involved in HNRNPL. Another convincing case is the transcription factor KLF5, which has been experimentally validated as a cancer driver gene in a recent publication[38]. This study reported three modes of action for KLF5 activation in cancer and noted that one of these functions was mediated by cancer type-specific mutations enriched in the zinc-finger motif. This finding is highly consistent with our prediction that the SPM E419Q within the zinc-finger motif may contribute to the oncogenic effect of KLF5, further demonstrating the effectiveness of our method. Finally, we found that the stop-gain SPM E225* in TFDP1 may explain the possible connection of this gene with tumorigenesis. TFDP1 can assemble into a protein complex by interacting with Rb and E2F proteins, which is necessary for the suppression of the cancer gene E2F through Rb protein[39]. When E2F is released, it positively regulates the progression of the cell cycle. The stop-gain mutation E225* in TFDP1 would delete the part that mediates interactions with both Rb and E2F (Fig. 5b) and might lead to the release of E2F. Thus, mutant TFDP1 could facilitate E2F release and promote cell cycle progression in the development of cancers.

## Discussion

By constructing a patient–mutation bipartite network across 33 TCGA cancer types, we systematically measured the cancer specificity of driver mutations and investigated the biological and clinical implications of this network. In contrast to previous studies[40,41] that mainly focused on cancer genes as a functional unit, we performed our analysis at the resolution of mutated residues to gain new insights of mutation-specific effects of cancer driver mutations. Comparing with Temko et al.'s work[20] that focused on why specific driver mutations were observed more

frequently in one cancer types, we associated the cancer specificity of mutations with potential functional effects and tumor behaviors. Our study is highly complemented with previous work on cancer driver specificity and represents a new angle to classify the functional consequences of gene mutations by leveraging rapidly accumulating sequencing data from diverse cancer types.

One of the most important observations from our study is that the co-occurrence of different types of driver mutations from the same cancer gene is a pervasive phenomenon. Half of the cancer genes in our study harbor both cancer type-specific and PCMs. Because driver mutations are supposed to confer selective advantages to tumor cells, the different cancer distribution patterns of driver mutations in one gene suggest that their capacities to promote tumor development are unequal[20]. Our observation further underscores the mutation-specific effects of driver mutations. Currently, the interpretation of many detected driver mutations in patients is usually transferred from studies on other cancer types[42]. We proposed that the applicability of the cross-cancer interpretation of driver mutations could be evaluated according to the position of this mutation in the cancer diversity spectrum. Specifically, the knowledge obtained from other caner types will be most useful for a PCM. However, for an SPM, this kind of knowledge will need to be carefully assessed.

We demonstrated the functional heterogeneity of mutations with different cancer diversity in a drug sensitivity analysis. For some tested drug–gene pairs, we showed that drug sensitivity is associated with the cancer diversity classification of driver mutations. Moreover, we found that a positive drug response often occurred in patients harboring SPMs. This finding is consistent with a new off-label clinical trial of the pan-HER kinase inhibitor neratinib[10], in which patients with breast cancer-specific mutations in the kinase domain showed a better neratinib response. Limited by available drug–gene pair data, we cannot directly evaluate our classification model on HER kinases. However, other important actionable genes, such as BRAF, EGFR, and KRAS, were tested and showed significant results. The influence of mutation-specific effects on targeted therapies may

**A**

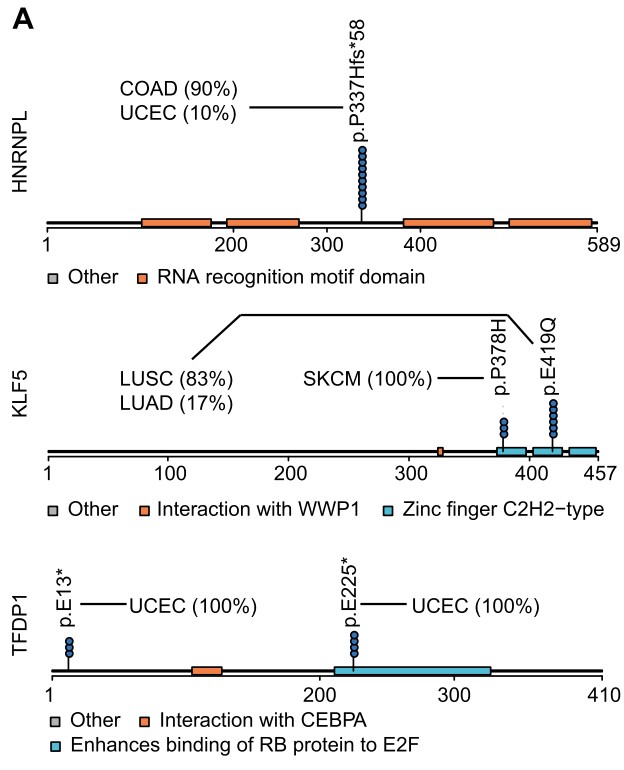

**B**

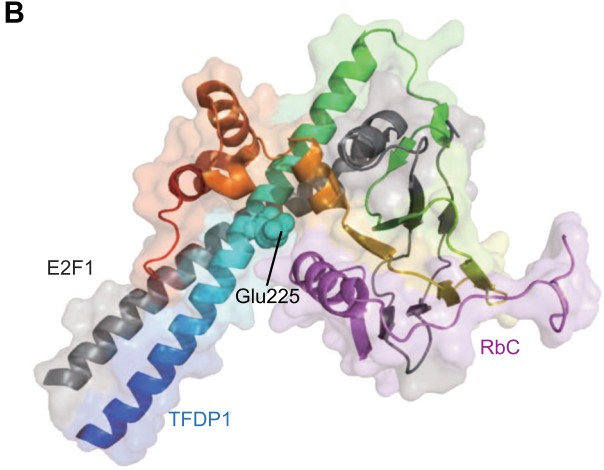

**Fig. 5 The new cancer genes identified by analyzing SPMs. a** The distribution of SPMs in functional protein domains. Each dot in the lollipop plot represents a patient. The pie plots show the cancer type composition of the patients. **b** The crystal structure of the protein complex comprised E2F1, TFDP1, and the Rb C-terminal domain (RbC). The PDB file was downloaded from the Protein Data Bank (https://www.rcsb.org/), which was identified as 2AZE. The structure was visualized with Pymol (https://pymol.org/2/) to highlight the 225th residue in TFDP1. RbC (light purple), E2F1 (gray), and TFDP1 (rainbow color, N-terminal to C-terminal is colored from blue to red).

be very common and should be considered in the design of off-label clinical trials. Our classification of driver mutations based on the cancer diversity spectrum could be an alternative reference to optimize patient enrollment conditions.

A number of factors could promote cancer evolution, including the persistent mutational process that generates new driver mutations, the genetic interactions between driver mutations, and the selective pressure from growing environment and therapy. To explore the association of cancer evolution and cancer diversity,

we analyzed the correlation of between clonal size and network diversity score, comutation patterns between driver mutations belonging to different cancer diversity categories and changes in network diversity values from primary to advanced tumors. The most prominent observation is that the cancer diversity of a driver mutation can change during cancer evolution, which to our knowledge, has not been characterized in previous related studies[40,41]. We found that many driver mutations lose their cancer type specificity in advanced stages. Although our study did not reveal the mechanism behind this phenomenon, several previous studies on the cancer transcriptome provide possible clues. For example, emerging evidence suggests that cancer development is an atavistic process. By comparing the expression profiles from different tumor cell types, it was shown that different tumors convergently evolve towards embryonic stem cell status[43]. The de-differentiation pattern of cells during disease progression has also been observed in MET500 project when comparing transcriptomes among normal, primary to metastatic samples[44]. Another study showed that unicell-origin genes are preferentially expressed in tumors and that coexpression between unicell-origin and multicell-origin genes is significantly lost[45]. Thus, we speculate that as cancer evolves and the atavistic process proceeds, the constraints from cell type-specific signaling may be gradually eliminated from tumors, resulting in the reduction of the cancer specificity of many driver mutations. However, it will be necessary to validate this speculation using in vitro or in vivo experiments in future work. Beyond that, there are many differences in experimental design and technologies used between TCGA and MSK-IMPACT, other explanations cannot be ruled out.

In conclusion, we employed a network framework to systematically measure the cancer distribution of driver mutations and successfully mapped them onto a cancer diversity spectrum. Even though there are some limitations in this work (see Supplementary Discussion for details), we have shown that this spectrum is informative in distinguishing the effects of drive mutations and hope that it could be a useful tool to improve the interpretation of detected driver mutations in clinical practice and to optimize off-label clinical trials in the personal genomic era.

## Methods

**Cancer mutation data and selection of cancer driver mutations.** Publicly available TCGA Mutation Annotation Format (MAF) files (GRCh38) for 33 cancer types were downloaded from the Genomic Data Commons (GDC) data portal (https://portal.gdc.cancer.gov/). Full names of cancer types can be found in Supplementary Data 13. These files contained 3,416,541 somatic mutations, including missense, silent, nonsense, splice site, frameshift insertions/deletions (indels), and inframe indels. The functional impact of each mutation was reannotated with M-CAP[46]. Because the scores of M-CAP were based on the genome version of GRCh37, we converted these scores into GRCh38 coordinates using liftOver, which was downloaded from USCS genome browser (http://genome.ucsc.edu/).

We extracted 616 cancer genes and information for their roles in cancer from COSMIC cancer gene census (v81). Cancer driver mutations were selected using the following criteria: First, all driver mutations must be from one of the 616 cancer genes. Second, we divided the cancer genes into three classes according to their roles in cancer, including OG, TSGs, and OG/TSG. Third, we selected recurrent OG mutations that had been recorded in COSMIC three times or more as drivers, and selected TSG mutations that would damage protein product functions as drivers. OG/TSG mutations matching one of these two requirements were selected. We defined truncated mutations and deleterious mutations predicted by M-CAP as damage mutations.

We predicted 40,358 driver mutations (approximately 1% of all mutations) for 454 cancer genes from 8326 patients (Supplementary Data 1). The average number of driver mutations harbored by a patient is 3.96, which is consistent with previous evaluation of the number of cancer drivers[23,41].

**Construction of the patient–mutation bipartite network and computation of network diversity scores.** We connected a patient and a driver mutation if this patient harbors the same mutation. This operation was performed in all selected TCGA mutations and resulted in a patient–mutation bipartite network. We used the following formula to compute the cancer distribution of a driver mutation,

named as the network diversity:

$$\text{network diversity}(m) = \frac{1}{\log k} \sum_{i=1}^{N} -p_i \log p_i, \qquad (1)$$

where $k$ is the degree of the mutation $m$ and $p_i$ represents the proportion of patients who harbor mutation $m$ from cancer type $i$. $N$ is the number of all cancer types in which $m$ is observed. The natural logarithm was used here. This formula actually calculates the entropy of the composition of cancer types, which is normalized by the number of observed patients. To balance the reliability and coverage of the measurement, only network diversity values of mutations observed in at least three TCGA patients were retained.

We refer to the information-theoretic analyses of Jenkinson et al.[47] on DNA methylation to partition the cancer diversity spectrum into three intervals. We defined an odds ratio of $r = p/1–p$. For one mutation that specifically mutates in a cancer type, the threshold is defined as $r > 10$ (i.e., the proportion of patients from a cancer type is 10 times or more than that of others), deriving $p > 0.9091$ or $p < 0.0909$. If we take all other cancer types as one type, then $r$ will correspond to $0 \leq$ network diversity $< 0.30$. In contrast, we defined $0.5 < r < 2$ (i.e., the proportion of patients from a cancer type is no more than 2 times that of others) as nonspecific (pancancer), resulting in $0.33 < p < 0.66$, which corresponds to $0.64 <$ network diversity $\leq 1$. Taken together, we split the cancer diversity spectrum into three parts: SPMs ($0 \leq$ network diversity $< 0.3$), RSMs ($0.3 \leq$ network diversity $< 0.64$), and PCMs ($0.64 \leq$ network diversity $\leq 1$).

**Functional analyses.** The genes enriched in each category of the cancer diversity spectrum were uploaded to STRING (https://string-db.org/) and expanded one time using a default neighboring algorithm, which added significantly connected genes into the current subnetwork. The enriched GO Biological Process terms were downloaded.

**Comutation analyses.** Fisher's exact test was used to analyze the significance of the overlap of patients between two drive mutations. The comutation pairs with $q < 0.1$ after Benjamini & Hochberg correction[48] were retained. We constructed 10,000 random comutation networks by shuffling the node labels, which could make every random network have the same degree distribution as the original network. Through random networks, we obtained an empirical distribution of the combination of mutations with different cancer diversity and derived a $p$ value for each of them. These $p$ values were also further corrected with the Benjamini & Hochberg method[48].

**Comparing the drug responses of patients harboring different types of cancer driver mutations.** We obtained an IDWAS data matrix from the supplemental materials in the study by Geeleher et al.[29]. The data generated by the general levels of drug sensitivity model were used. Furthermore, we downloaded the gene–drug interaction data from DGIdb[49], which indicates whether a mutated gene can be targeted by a specific drug. The efficacy of the drug–gene pair was tested as described below. For a specific driver gene, we split patients who harbor mutations of the driver gene into three tiers according to the cancer diversity type of mutations (i.e., SPM, RSM, or PCM). Then, we evaluated the drug responses associated with patient tiers for a given potential drug using ANOVA and compared the drug responses of each tier with those of mutation-negative patients using a two-sided $t$-test. To satisfy the requirement of statistical testing, we limited our analysis to driver genes with three or more patients in every tier.

Clinical drug recommendation information was downloaded from the OncoKB database (http://oncokb.org/), and we precisely mapped driver mutations to OncoKB actionable mutations and drug annotations.

**Cancer evolution analyses.** We analyzed the correlation between mutation VAF and network diversity values. To exclude the influence of tumor purity and copy number variation (CNV) on VAF values, we first downloaded tumor purity data estimated by InfiniumPurify[50], which used DNA methylation to systematically analyze the purity of samples from 32 TCGA cancer types. Samples with tumor purity <0.7 were ignored in our study. Level 3 CNV data generated by Affymetrix SNP 6.0 array were downloaded from the GDC data portal (https://portal.gdc.cancer.gov/). Regions with an absolute score ≤0.5 were deemed CNV neutral. We only used VAF values of mutations in CNV neutral regions and highly pure tumor samples to compute the correlation between VAF and network diversity values. Pearson's correlation coefficient and related statistical tests were performed using R language.

For comparison between primary and advanced tumors, we obtained the targeted sequencing data of advanced tumors generated by MSK-IMPACT from the cBioPortal (http://www.cbioportal.org/). We computed network diversity values for all mutations with recurrent frequency ≥3 in MSK-IMPACT data. The network diversity distributions of 625 overlapping mutations between TCGA and MSK-IMPACT were compared.

**Identification of new driver genes.** We computed network diversity values for all mutations that occur three times or more in 33 types of cancer reserved in TCGA

and all SPMs were collected. The MutSigCV module in GenePattern[51] was used with default settings to search for genes significantly enriched with SPMs ($q < 0.001$). We downloaded cancer dependency data from https://depmap.org/rnai/ and intersected 140 unrecorded candidate genes with 769 cancer-dependent genes to obtain eight new cancer genes. The impact of the expression of these eight genes expression on patient survival was analyzed by querying the Human Pathology Atlas (http://www.proteinatlas.org/pathology)[52].

**Network data manipulation and visualization.** Network data were computed using the NetworkX package (https://networkx.github.io) and further visualized in Cytoscape[53]. The firework web portal was built on the Perl-based web framework, "Catalyst".

**Statistics and reproducibility.** Statistical analyses and FDR correction were performed with R language (v3.4.1). The methods used to perform statistical test for each analysis can be found in above sections.

**Reporting summary.** Further information on research design is available in the Nature Research Reporting Summary linked to this article.

## Data availability

We only used publicly available data. The accession codes of TCGA data can be found in Supplementary Data 14. Cancer diversity of driver mutations can be visualized at http://mulinlab.org/firework. User can query each mutation to check its network diversity and its distribution in 33 cancer types. The position for mutations from one gene will be shown as lollipop plot. The drug information for studied mutations can be found in Supplementary Data 8 and Table 9. And the full patient–mutation network can also be found in Supplementary Data 15.

## Code availability

The computer code for network diversity calculation and other codes used in this work are available from corresponding authors on request.

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

## Acknowledgements

We acknowledge the members of the Prof. Guang Li Labs for critical discussions. We also appreciate all tool and resource providers. This work was supported by grants from the National Natural Science Foundation of China 31801122 (to X.D.), 31701143, 31871327 (to M.J.L.), and Natural Science Foundation of Tianjin 18JCZDJC34700, 19JCJQJC63600 (to M.J.L.).

## Author contributions

X.D. conducted the computational analysis described in the manuscript. D.H. implemented the web portal. X.D. and M.J.L. conceived the study and wrote the paper. X.Y., S.Z., and Z.W. evaluated the computational methods, tested the web portal, and reviewed the manuscript. B.Y., P.C.S., and K.C. contributed to co-ordination and supervision of computational methodologies, and reviewed the manuscript. All authors read and approved the final manuscript.

## Competing interests

The authors declare no competing interests.
