## [Peer Review File · Communications Biology]

Reviewers' comments:

Reviewer #1 (Remarks to the Author):

Dong et al. used TCGA data on 33 cancer types to construct a patient-mutation network and identified mutations that are cancer-specific, relatively specific and pancancer. Interestingly, many genes harbor mutations that are associated with different classes, suggesting that, even if different mutations affect the same domain, they may have different effects and be associated with different tumor types. These differences seem to reflect different responses to anti-cancer drugs and may be used to select the most appropriate treatment.

Overall, the authors used a very-well characterized dataset, such as TCGA, and integrated it with multiple publicly available sources to describe driver mutations from a new perspective that may result in better treatment for cancer patients.

Major comments:

In the abstract, "allele-specific effects" are introduced in the first sentence, but never mentioned again: the authors should describe why/how they use allele-specific effects in the context of this study. The fact that in the manuscript the authors describe "variant allele frequency" makes the concept more confusing. I would suggest to use one consistent term in the manuscript.

Line 201: "driver genes from these 202 three categories are highly overlapped. A total of 18% of genes with mutations were 203 distributed in all three categories, and 50% were distributed in at least two categories". How is it possible that a gene belongs to multiple classes? The authors performed this analysis mutation by mutation: please, rephrase this sentence to be less confusing.

Line 209: cancer predispositions should be introduced better: why did the authors choose to investigate it? Also, the next analysis described in the manuscript (GO enrichment) is not related to cancer predisposition. I would switch the order of these two paragraphs and move cancer predisposition to a section by itself.

Line 308-310: the authors should test the differences between oncogenes and tumor suppressors. Looking at figure 4B, there doesn't seem to be a strong association.

Line 376: 4 of the 8 novel bona fide cancer genes are included in the Network of Cancer Genes (NCG: <http://ncg.kcl.ac.uk>), confirming that they are known to be likely driver. Please, acknowledge this fact.

Minor comments:

Please provide a supplemental table with the abbreviations for all cancer types

P-values in Table S8 should be FDR-corrected

Some references in the methods are formatted in different ways.

Matteo D'Antonio

Reviewer #2 (Remarks to the Author):

Dong et al. developed a network-based framework to systematically measure cancer diversity for each driver mutation. They performed their analysis at resolution of mutated residues resolution to gain some new insights about allele-specific effects of driver mutations. By projecting mutations into a cancer diversity spectrum, the authors classified these mutations into three categories, including cancer-specific, relatively specific and pancancer mutations through projecting mutations onto a cancer diversity spectrum. They systematically investigated the distribution of these mutations in protein domains, genes, and cellular pathways as well as their co-mutation patterns. To demonstrate the potential value of the cancer diversity spectrum for clinical and biological significance, they leverage this information to predict patient drug responses and identify new cancer driver genes. The authors finally developed a website to present the cancer diversity for driver mutations. I have several comments on this manuscript below:

1. What is the force-directed layout algorithm? A brief description may help readers better understand it.
2. The proposed approach perhaps works better for identifying gain-of-function driver mutations (i.e., oncogenes) in specific protein regions, whereas loss-of-function driver mutations (i.e., tumor suppressors) resulting randomly from truncated mutations may be missed. This should be clarified in the paper.
3. It would be great if the authors add some in vitro and in vivo experiments to validate their findings.
4. Apart from the Pearson correlation coefficient, the concordance correlation coefficient could be evaluated, too.
5. The reference of Benjamini & Hochberg correction should be added.
6. In line 836(Figure legend of 3B), **: $p < 0.01$.

Reviewer #3 (Remarks to the Author):

This paper presents a pilot study on cancer distribution of driver mutations using TCGA. The authors developed a network-based framework based on cancer diversity for each driver mutation. The major findings include 1) co-occurrence of different types of driver mutations in same cancer gene. 2) The specificity of the mutations could influence patient drug responses. Moreover, increased diversity was observed in advanced tumors. 3) The authors discovered some potentially novel cancer driver genes based on the diversity spectrum. In summary, the diversity spectrum analysis is useful to define driver mutations and optimize clinical trials.

I think this paper is novel and will be of interest to others in the community. The work is convincing except some minor comments I listed in the attachment.

The statistical part is valid and makes sense. To facilitate other researchers, the authors also provide a website portal to visualize any mutation of interest.

Response to the comments of Review #1:

Reviewer #1: Dong et al. used TCGA data on 33 cancer types to construct a patient-mutation network and identified mutations that are cancer-specific, relatively specific and pancancer. Interestingly, many genes harbor mutations that are associated with different classes, suggesting that, even if different mutations affect the same domain, they may have different effects and be associated with different tumor types. These differences seem to reflect different responses to anti-cancer drugs and may be used to select the most appropriate treatment.

Overall, the authors used a very-well characterized dataset, such as TCGA, and integrated it with multiple publicly available sources to describe driver mutations from a new perspective that may result in better treatment for cancer patients.

Major comments:

1. In the abstract, “allele-specific effects” are introduced in the first sentence, but never mentioned again: the authors should describe why/how they use allele-specific effects in the context of this study. The fact that in the manuscript the authors describe “variant allele frequency” makes the concept more confusing. I would suggest to use one consistent term in the manuscript.

Response: We thank the reviewer for the constructive suggestion, which reminds us that our readers might feel confused with the words “allele” and “mutation” in our manuscript. Accordingly, we have changed all “allele” to “mutation” in our revised manuscript (except for “variant allele frequency”) and rephrased the title of our manuscript as “Diversity spectrum analysis identifies mutation-specific effects of cancer driver genes”.

2. Line 201: “driver genes from these three categories are highly overlapped. A total of 18% of genes with mutations were distributed in all three categories, and 50% were distributed in at least two categories”. How is it possible that a gene belongs to multiple classes? The authors performed this analysis mutation by mutation: please, rephrase this sentence to be less confusing.

Response: We apologize for the confusion again. The reviewer is right that we classified mutations one by one then analyzed corresponding driver genes in each category. Thus, a gene could be associated with multiple categories of driver mutation. In the manuscript, we wanted to express that mutations belong to different categories can be observed in one driver gene.

We have rephrased these sentences as “Driver genes that harbor multiple types of mutations are common. A total of 18% of genes harbor three types of mutations and 50% of genes harbor at least two types of mutations (Figure 2C).” (lines 231-233)

3. Line 209: cancer predispositions should be introduced better: why did the authors choose to investigate it? Also, the next analysis described in the manuscript (GO enrichment) is not related to cancer predisposition. I would switch the order of these two

paragraphs and move cancer predisposition to a section by itself.

Response: We thank the review for the comments.

Patients with hereditary cancer predisposition syndromes suffer from high cancer risk since they carry germline cancer driver mutations. Even these mutations exist in all tissue cells of the whole body, the risks of tumorigenesis vary with tissue types and mutated genes, which could provide an orthogonal validation of the cancer mutation specificity learned from TCGA somatic mutation data using ND measurement. Thus we analyzed the cancer predispositions of genes enriched in different cancer diversity spectrum categories. According to the suggestion of the reviewer, we have added some words to introduce our intention and used an individual section to describe this result. (lines 253-274)

Also, we have moved the paragraph describing pathway analysis to front of hereditary cancer predisposition analysis section. (lines 239-251)

4. Line 308-310: the authors should test the differences between oncogenes and tumor suppressors. Looking at figure 4B, there doesn't seem to be a strong association.

Response: Thank you for the review. The average Pearson correlation coefficient between NDs and VAFs for 25 oncogenes and 25 tumor suppressor genes in Figure 4B are -0.08 and 0.01 respectively. Wilcoxon sum rank test shows the difference between them is not significant (p -value= 0.079). These data have been included in our manuscript. We also rephrased our conclusion in this paragraph as

“Considering that high VAF generally indicates an early tumor clone, our results imply that a part of oncogene-related SP mutations and tumor suppressor-related PC mutations tend to occur in the early stage of tumorigenesis.”

5. Line 376: 4 of the 8 novel bona fide cancer genes are included in the Network of Cancer Genes (NCG: <http://ncg.kcl.ac.uk>), confirming that they are known to be likely driver. Please, acknowledge this fact.

Response: We are grateful to the reviewer for giving this important information. We have added this validation in revised manuscript (lines 423-426).

Minor comments:

1. Please provide a supplemental table with the abbreviations for all cancer types

Response: According to the suggestion, we have given all full names and abbreviations of all cancer types in Table S13 and Figure 1 legend.

2. P-values in Table S8 should be FDR-corrected

Response: FDR values calculated through Benjamini & Hochberg correction method have been added in Table S8.

3. Some references in the methods are formatted in different ways.

Response: Corrected.

Response to the comments of Review #2:

Reviewer #2: Dong et al. developed a network-based framework to systematically measure cancer diversity for each driver mutation. They performed their analysis at resolution of mutated residues resolution to gain some new insights about allele-specific effects of driver mutations. By projecting mutations into a cancer diversity spectrum, the authors classified these mutations into three categories, including cancer-specific, relatively specific and pancancer mutations through projecting mutations onto a cancer diversity spectrum. They systematically investigated the distribution of these mutations in protein domains, genes, and cellular pathways as well as their co-mutation patterns. To demonstrate the potential value of the cancer diversity spectrum for clinical and biological significance, they leverage this information to predict patient drug responses and identify new cancer driver genes. The authors finally developed a website to present the cancer diversity for driver mutations. I have several comments on this manuscript below:

1. What is the force-directed layout algorithm? A brief description may help readers better understand it.

Response: We thank the reviewer for the good suggestion. Force-directed layout algorithm is an intuitive method to spatially organize network data within, usually, a two-dimensional plane. Nodes in the network will repel each other as they were like charged bubbles. On the other hand, each edge will act like a spring to pull a pair of connected nodes together. As the result, cancer types associated with similar driver mutation sets will be clustered and pushed away from other cancer types with different mutation profiles in the final network (Figure 2A), which allows us to observe the similarity among these cancer types in a globally and flexible manner.

We have added the description of force-directed layout algorithm and a related reference to our manuscript. (lines 171-180)

2. The proposed approach perhaps works better for identifying gain-of-function driver mutations (i.e., oncogenes) in specific protein regions, whereas loss-of-function driver mutations (i.e., tumor suppressors) resulting randomly from truncated mutations may be missed. This should be clarified in the paper.

Response: We appreciate the reviewer for pointing this out. We agree with reviewer that our approach is better for identifying gain-of-function driver mutations for now. The main requirement of our approach is the recurrence (≥ 3 times in the work) of a mutation in cancers but not the mutation type (i.e. gain-of-function or loss-of-function). If a loss-of-function (such as a frameshift or truncated mutation of a tumor suppressor) is observed many times in cancer, we can measure its ND value, such as *APC* Q1291*, *NPM1* W288Cfs*12 and *TP53* R175H in this work. On the contrary, our approach does not work on a very rare gain-of-function driver mutation since the computing of ND is impossible or the value is unreliable in this circumstance. Nevertheless, considering that gain-of-function mutations tend to occur at a specific position and the loss-of-function mutations occur relatively randomly, the recurrence ratio of gain-of-function mutations would be higher and more of them would satisfy the requirement of our approach. We

think a possible solution for this shortcoming would be clustering some very similar loss-of-function mutations before performing cancer diversity analysis, which can improve the recurrence ratio of them.

And we thank the reviewer again for this constructive comment. We have discussed this problem in our manuscript as:

“Third, our approach requires that a mutation is recurrent in cancers, otherwise, the computing of ND would be impossible or the value would be unreliable. Considering that gain-of-function mutations tend to occur at a specific position and the loss-of-function mutations occur relatively randomly on a gene, the recurrence rate of gain-of-function mutations would be higher and more of them would satisfy the requirement of our approach but some loss-of-function mutations would be missed. We think a possible solution for this shortcoming would be clustering some very similar loss-of-function mutations before performing cancer diversity analysis, which can improve the recurrence rate of them.” (lines 525-533)

3. It would be great if the authors add some in vitro and in vivo experiments to validate their findings.

Response: We thank the reviewer for this suggestion.

We have actually considered doing some experiments to test the functions of the eight newly identified potential cancer driver genes using cancer cell lines. But we noticed the dependency of these candidate driver genes to cancer cells already had been experimentally tested using RNAi approach by Tsherniak et al. (Tsherniak, A., et al. (2017). "Defining a cancer dependency map." *Cell* 170(3): 564-576.e516.).

And Reviewer #1 has also pointed out that the four of the eight potential cancer driver genes have been included in the latest version of The Network of Cancer Genes, a manually curated cancer gene database. We also found three additional identified potential driver genes have been incorporated into CancerMine database, which indicates most of our predicted cancer driver genes are discovered and validated recently.

Thus, we decided not to further verify their functions systematically using experimental approaches. However, we acknowledge the importance of experimental validation for genomic analysis and will further validate our findings in future work.

4. Apart from the Pearson correlation coefficient, the concordance correlation coefficient could be evaluated, too.

Response: We thank the review for this suggestion. We have calculated concordance correlation coefficients (CCC) ¹ between ND and VAF of each genes using the `epi.ccc` function from R package `epiR`. Generally, the Pearson correlation coefficients (PCCs) and CCCs are highly correlated (Figure R1). Note that the CCC values are smaller than PCC values. Because CCC measures the agreement between ND and VAF. A high CCC value means the points in scatterplot should distributed around the 45 degrees line. For our analysis, we only want to measure if there is a linear relationship between VAF and ND and the points can distribute around any straight lines (except for lines paralleling with x or y axis). Thus, PCC is more suitable.

Figure R1. The correlation of Pearson correlation coefficient (PCC) and concordance correlation coefficient (CCC) in the measurement of the relationships between ND and VAF of cancer genes.

5. The reference of Benjamini & Hochberg correction should be added.

Response: We thank the reviewer for pointing this out. The reference (Ref. 49) has now been added in the revised manuscript.

6. In line 836 (Figure legend of 3B), **: $p < 0.01$.

Response: It has been corrected as "The red stars mark statistically significant groups when compared with corresponding negative groups (*: $p < 0.05$, **: $p < 0.01$, two-sided t -test)."

Reference

1. Lawrence IKL. A concordance correlation coefficient to evaluate reproducibility. *Biometrics* 45, 255-268 (1989).

Response to the comments of Review #3:

Reviewer #3: This paper presents a pilot study on cancer distribution of driver mutations using TCGA. The authors developed a network-based framework based on cancer diversity for each driver mutation. The major findings include 1) co-occurrence of different types of driver mutations in same cancer gene. 2) The specificity of the mutations could influence patient drug responses. Moreover, increased diversity was observed in advanced tumors. 3) The authors discovered some potentially novel cancer driver genes based on the diversity spectrum. In summary, the diversity spectrum analysis is useful to define driver mutations and optimize clinical trials.

I think this paper is novel and will be of interest to others in the community. The work is convincing except some minor comments I listed in the attachment.

The statistical part is valid and makes sense. To facilitate other researchers, the authors also provide a website portal to visualize any mutation of interest.

Major Comments:

1. In line 152, could the authors please clarify how you removed the mutations that occur less than three times? The removal is on per cancer type level? Or on all cancer types in TCGA as a whole?

Response: We apologize for the confusion. We removed the mutations that occur less than three times on the whole TCGA dataset. We have clarified this in the revised manuscript (lines 163-165).

2. Did the authors use MAF format mutation data? If so, did you filter out the mutation with low impact?

Response: Yes, we used MAF files downloaded from the Genomic Data Commons (GDC) data portal (<https://portal.gdc.cancer.gov/>) and filtered out low impact mutations. We conducted filtering by selecting driver mutations through combing the role of corresponding cancer genes (oncogene/tumor suppressor gene) and the pathogenic scores of mutations. The detailed description of this operation can be found in first section of Methods.

To improve the readability of our manuscript, we moved the explanation of the rationale of driver mutation selection to the first section of Results (lines 149-159).

3. In line 177, the authors explained the meaning of ND value that higher value indicates the mutation is observed in multiple cancer types with a more similar possibility. So I'm guessing the lower value indicates fewer cancer types with more different possibilities? How about the mutation occurs in multiple cancer types with more different possibilities? I think the authors should give a more detailed explanation of ND value with diverse scenarios.

Response: We thank the reviewer for the comments. Generally, the reviewer is right that if a mutation occurs in fewer cancer types in which one of them has a very high possibility of being observed and the possibilities for the other cancer types are low, the ND value will be low. As an entropy-based approach, ND measures whether a mutation observed in different cancer types with similar probability, but not the number of cancer types associated with it. If a mutation occurs in multiple cancer types and a cancer type dominates the cancer type composition, the ND value will be low. On the contrary, if the possibilities of multiple cancer types are similar, the ND value will be high. For example, although both *KRAS* G12V and *KRAS* G12R occur in > 5 different cancer types, their probabilistic distributions of cancer types are different. There are total 37 patients associated *KRAS* G12R in our data. Above 75% of them are PADD patients. In contrast,

for the 176 patients associated with *KRAS* G12V, there are three cancer types occupy much of the composition (23% of PADD, 22% of LUAD and 19% of COAD). Thus, the ND value of it (G12V, ND=0.40) is relatively high than *KRAS* G12R (ND=0.28), representing a different cancer specificity. We have added this explanation in the new manuscript (lines 198-207).

4. I noticed PTEN appears to be in specific mutations group (line 190) as well as relatively specific mutations group (line 197). I recommend to clarify the cancer types in two groups, like how the authors demonstrate germline mutations in a very clear manner. Similarly, TP53 is in both RS (line 197) and PC (line 198).

Response: We thank the reviewer for the suggestions. Because the cancer types associated with each mutation can be different even for the same gene. If we list all cancer types for each mutation of genes, it will be a very lengthy sentence and very difficult to read. We think the best way to check the cancer types of each mutation is using our web portal, in which the interactive graphs provide the classification of each mutation of a gene and detailed cancer types associated with them. We have given this suggestion to our reader in the new manuscript (lines 237-238).

5. The author mentioned that driver genes are highly overlapped in three categories. Followed by that, authors should explain the reason to make it clear. I think a better way is to do the classification on the allele specific mutation level rather than gene level? This way there will be no overlapping issues which is a little bit confusing.

Response: We thank the reviewer for the comment. We apologize for the confusion. We performed classification on the mutation level then analyzed corresponding driver genes in each mutation category. A gene could be associated with many mutations. And thus mutation can belong to different mutation categories. Thus, a gene could be associated with multiple mutation categories.

Please also see our response to the second comment from Reivewer #1, and we have rephrased related sentences in our revised manuscript (lines 231-233).

Minor Comments:

1. In line 190 - 193, it is better to label the specific cancer for each specific mutation, since readers could be interested and eager to know the pairs of mutation and cancer type and existing drugs developed based on that. These findings may also serve as a “proof-of-concept” to demonstrate your method actually works.

Response: We thank the review for the good suggestion. This part has been rewritten as “This category also includes many known biomarkers for cancer diagnosis or targeted treatment, such as *APC* Q1291* (for COAD), *EGFR* L858R (for LUAD), *BRAF* V600E (for THCA and SKCM), *DNMT3A* R882H and *NPM1* W288Cfs*12 (for LAML).” (lines 220-223)

2. It will be easier for the readers if the authors could label Figure 2D-F with the names of the categories.

Response: We thank the review for the suggestion. We have added the abbreviations of

three categories to the right side of barplots.

3. Is there a specific reason why the dash line doesn't pass through the first two pathways in Figure 2D?

Response: We apologize for the confusion. The meaning of dash line in Figure 2D is same as in Figure 2E and Figure 2F. We have corrected the length of this dash line.

4. In paragraph from line 274, it is necessary to clarify what the imputed drug response stands for (IC50 thing), and explain what the negative/positive value stands for.

Response: Because the drug response data from IDWAS are predicted from a gene expression-based statistical model, the drug response values from IDWAS have no clearly defined biological meaning and are not directly comparable with traditional drug sensitivity values such as IC50, however, lower value means greater drug sensitivity. We have added these explanations to our new manuscript (lines 313-318).

5. The colors in Figure 3D are really hard to distinguish. The authors may use more differentiated colors.

Response: Thank you for the suggestion. We have adjusted the colors used in Figure 3D to make it more clear.

6. In Figure 4B, the colors in legend (brighter) don't match with the colors in the figure (darker). The Y axis should be $-\log_{10}(\text{P-value})$?

Response: We thank the reviewer for pointing this out. We have adjusted the legend to keep it consistent with colors of dots in the figure. And the " $\log_{10}(\text{p-value})$ " has also been corrected as " $-\log_{10}(\text{p-value})$ ".

7. In the Firework web portal for the mutation lollipop plot, I recommend the authors to change the shape of the legend label from square to round shape. Since the genes are in rectangle shape and mutations are labelled as a circle.

Response: We thank the reviewer for pointing this out. We have changed the shape of legend in our web portal.

REVIEWERS' COMMENTS:

Reviewer #1 (Remarks to the Author):

The Authors adequately responded to all my comments.

Reviewer #2 (Remarks to the Author):

The authors did a good job and have largely addressed my comments. I do not have further comments